# Venous Thromboembolism and Bleeding after Transurethral Resection of the Prostate (TURP) in Patients with Preoperative Antithrombotic Therapy: A Single-Center Study from a Tertiary Hospital in China

**DOI:** 10.3390/jcm12020417

**Published:** 2023-01-04

**Authors:** Zhongyi Li, Zhihuan Zheng, Xuesong Liu, Quan Zhu, Kaixuan Li, Li Huang, Zhao Wang, Zhengyan Tang

**Affiliations:** 1Department of Urology, Xiangya Hospital, Central South University, Changsha 410008, China; 2Department of Critical Care Medicine, Xiangya Hospital, Central South University, Changsha 410008, China

**Keywords:** antithrombotic therapy, transurethral resection of the prostate, postoperative hemorrhage, venous thromboembolism

## Abstract

Background: Venous thromboembolism (VTE) and postoperative hemorrhage are unavoidable complications of transurethral resection of the prostate (TURP). At present, more and more patients with benign prostate hyperplasia (BPH) need long-term antithrombotic therapy before operation due to cardiovascular diseases or cerebrovascular diseases. The purpose of this study was to investigate the effect of preoperative antithrombotic therapy history on lower extremity VTE and bleeding after TURP. Methods: Patients who underwent TURP in the Department of Urology, Xiangya Hospital, Central South University, from January 2017 to December 2021 and took antithrombotic drugs before operation were retrospectively analyzed. The baseline data of patients were collected, including age, prostate volume, preoperative International Prostate Symptom Score (IPSS), complications, surgical history within one month, indications of preoperative antithrombotic drugs, drug types, medication duration, etc. Main outcome measures included venous thromboembolism after TURP, intraoperative and postoperative bleeding, and perioperative blood transfusion. Secondary outcome measures included operation duration and postoperative hospitalization days, the duration of stopping antithrombotic drugs before operation, the recovery time of antithrombotic drugs after operation, the condition of lower limbs within 3 months after operation, major adverse cardiac events (MACEs), and cerebrovascular complications and death. Results: A total of 31 patients after TURP with a long preoperative history of antithrombotic drugs were included in this study. Six patients (19.4%) developed superficial venous thrombosis (SVT) postoperatively. Four of these patients progressed to deep vein thrombosis (DVT) without pulmonary thromboembolism (PE). Only one patient underwent extra bladder irrigation due to blockage of their urinary catheter by a blood clot postoperatively. The symptoms of hematuria mostly disappeared within one month postoperatively and lasted for up to three months postoperatively. No blood transfusion, surgical intervention to stop bleeding, lower limb discomfort such as swelling, MACEs, cerebrovascular complications, or death occurred in all patients within three months after surgery. Conclusion: Short-term preoperative discontinuation may help patients with antithrombotic therapy to obtain a relatively safe opportunity for TURP surgery after professional evaluation of perioperative conditions. The risks of perioperative bleeding, VTE, and serious cardiovascular and cerebrovascular complications are relatively controllable. It is essential for urologists to pay more attention to the perioperative management of these patients. However, further high-quality research results are needed for more powerful verification.

## 1. Introduction

Benign prostate hyperplasia (BPH) is a common urinary system disease in elderly men which greatly affects the quality of life for these patients. The clinical manifestations are mainly lower urinary tract symptoms (LUTS) such as frequent micturition, urgency, increased nocturia, weak micturition, incomplete urination, etc. Transurethral resection of the prostate (TURP) is the main surgical method for BPH patients, and its risks include postoperative bleeding and venous thromboembolism (VTE) [1,2,3,4]. VTE refers to deep vein thrombosis (DVT) and pulmonary thromboembolism (PE) [5]. However, superficial venous thrombosis (SVT) is more common in clinical practice [6].

The incidence of BPH increases with age. Many elderly patients need long-term antithrombotic therapy before surgery due to cardiovascular or cerebrovascular diseases [7]. Antithrombotic therapy includes anticoagulation and antiplatelet therapy. If antithrombotic therapy is discontinued during the perioperative period, the risk of cardiovascular and cerebrovascular events will increase [8], while continuing therapy will increase the risk of bleeding after TURP [9]. A previous study showed that the history of antithrombotic drug treatment within one month was an independent risk factor for VTE after urological non-malignant tumor surgery, and the risk of VTE after surgery was markedly increased 10-fold compared to that of patients without antithrombotic drug use [10]. The timing of preoperative discontinuation of antithrombotic drugs is critical. However, the existing studies [11,12] mostly focus on the influence of stopping time on postoperative hemorrhage risk and fail to comprehensively assess the risk of hemorrhage and VTE. In addition, most studies [13,14,15] only analyzed the effect of aspirin on TURP surgery, which is difficult to fully adapt to the actual complex clinical situation. Therefore, the purpose of this study is to explain the occurrence of VTE and bleeding after TURP in patients who stopped using antithrombotic drugs before operation, which may provide a reference for clinical practice.

## 2. Materials and Methods

### 2.1. Study Population

This study is a retrospective study and has been approved by the Ethics Committee of Xiangya Hospital of Central South University (No. 202011183). The inclusion criteria were as follows: (1) patients who underwent TURP surgery in the Department of Urology, Xiangya Hospital, Central South University, from January 2017 to December 2021; (2) duration for maintenance antithrombotic drugs before operation of more than one month. The following were exclusion criteria: (1) patients complicated with active malignant diseases; (2) a history of prostate surgery or urinary tract reconstruction surgery; (3) postoperative pathological examination showing prostate cancer; (4) VTE detected preoperatively; (5) bridging therapy such as low-molecular-weight heparin before operation.

According to the epidemiology and clinical symptoms of BPH patients, referring to the guidelines and experts’ consensus on the prevention and treatment of BPH and VTE, the clinical data we collected mainly included age, prostate volume, preoperative International Prostate Symptom Score (IPSS), complications (hypertension, coronary heart disease, diabetes, stroke, varicose veins of lower limbs, etc.), surgical history within one month, and preoperative use of antithrombotic drugs.

### 2.2. Outcome Measures

The main outcomes are venous thromboembolism after TURP, intraoperative and postoperative bleeding, and perioperative blood transfusion. The amount of bleeding and the duration of the operation are all referred to in the surgical anesthesia record sheet. Postoperative bleeding can be divided into whether there is slight gross hematuria (reddish) or obvious gross hematuria (crimson) within 3 months after operation and whether extra bladder irrigation or re-operation is needed. Secondary outcomes include operation duration and postoperative hospitalization days, the duration of stopping antithrombotic drugs before operation, the recovery time of antithrombotic drugs after operation, the condition of the lower limbs, major adverse cardiac events (MACEs), cerebrovascular complications, and death within 3 months after operation. MACEs mainly mean acute myocardial infarction, cardiac arrest, severe arrhythmia, cardiac death, etc. Cerebrovascular complications mainly refer to stroke and transient ischemic attack.

### 2.3. Perioperative Management

In order to control the risk of perioperative bleeding, the patients who maintained antithrombotic therapy stopped taking antithrombotic drugs before TURP after evaluation and guidance by relevant specialists, and none of them used bridging therapy. After the risk of postoperative bleeding decreased, the original antithrombotic protocol was reactivated.

All patients were treated with mechanical thromboprophylaxis to prevent venous thromboembolism during the perioperative period. On the surgery day, patients were guided by specialized nurses to wear appropriate graduated compression stockings (GCS). The frequency of removing the graduated compression stockings was limited to three times a day, and the duration of removing time was limited to half an hour. Intermittent pneumatic compression (IPC) was applied after the postoperative patient returned to the ward [16].

Before and after the operation, the patients were examined by ultrasound in both lower limbs, which was performed by experienced sonographers. If a patient has postoperative symptoms such as dyspnea, syncope, hemoptysis, chest pain, shock, or decreased oxygen saturation, further examination such as pulmonary CTA and/or echocardiography is required to determine the occurrence of PE.

Once the patient was diagnosed with VTE or SVT, mechanical prevention of thrombosis (GCS and IPC) was immediately stopped according to the consultation opinion from the VTE group, and the risk of bleeding was assessed before anticoagulant therapy or even thrombolytic therapy immediately under the guidance of the VTE professional team.

### 2.4. Follow-Up

Follow-up was carried out at 7 days, 1 month, and 3 months after operation, mainly through outpatient service and telephone calls. The follow-up included drugs and the duration of prescription, hematuria or bleeding 3 months after surgery, lower extremity conditions, MACEs, cerebrovascular complications, and death.

### 2.5. Statistical Analysis

Data were analyzed using IBM SPSS 26.0 software. Quantitative data conforming to the normal distribution are expressed as mean ± standard deviation ($\bar{x}$ ± s), while those not conforming to the normal distribution are expressed as median (interquartile range). Qualitative data are expressed as cases (percentage) (n(%)).

## 3. Result

### 3.1. Patients’ Baseline Data

A total of 31 patients who underwent TURP with a long history of taking antithrombotic drugs before operation (Table 1) were included in this study. The mean age of the 31 patients was 70.3 ± 6.5 years, the mean preoperative IPSS score was 20.2 ± 3.0, and the median prostate volume was 56.2 (44.9–85.6) mL. One patient (3.2%) underwent prostate biopsy within one month before TURP.

Among the 31 patients, 6 (19.4%) had coronary stent implantation, 2 (6.5%) had aortic valve replacement, 3 (9.7%) had mitral valve replacement, 2 (6.5%) had a history of myocardial infarction, 10 (32.3%) had a history of cerebral infarction, 6 (19.4%) had a history of coronary heart disease, and 2 (6.5%) had a history of atrial fibrillation. Two patients (6.5%) took long-term oral warfarin and twenty-nine patients (93.5%) took long-term oral antiplatelet drugs, including 19 patients (61.3%) taking aspirin, 6 patients (19.4%) taking clopidogrel, and 4 patients (12.9%) taking aspirin combined with clopidogrel. Three patients (9.7%) took medicine for less than one year, fifteen patients (48.4%) for 1–5 years, eleven patients (35.5%) for 5–10 years, and two patients (6.5%) for more than 10 years.

### 3.2. Incidence of VTE after TURP

SVT after surgery occurred in 6 (19.4%) of 31 TURP patients with a history of taking antithrombotic drugs preoperatively. Among them, four patients developed DVT without PE. In the remaining 25 patients, there was no SVT/DVT/PE (Table 2). 

### 3.3. Perioperative Situation of TURP

Among the 31 patients, antithrombotic drug discontinuation occurred in three cases (9.7%) within one week before surgery. In 24 cases (77.4%), the withdrawal time span was between one and two weeks, and in the other four cases (12.9%), it was more than two weeks. After preoperative drug discontinuation in 31 patients, no bridging therapy was performed and no new adverse events such as myocardial infarction and cerebral infarction occurred. The median operation duration was 75 (50–100) min, and the median intraoperative bleeding volume was 30 (10–100) mL. All patients were discharged 2–4 days after surgery, and nobody needed a blood transfusion during hospitalization (Table 3). 

All patients were followed up at 7 days, 1 month, and 3 months after surgery. During the follow-up period, there was no lower extremity discomfort such as swelling, no MACEs, no cerebrovascular complications, and no death among all patients (Table 3). Two patients (6.5%) resumed antithrombotic therapy within 1 week after surgery. In 27 patients, the time for returning to antithrombotic drugs was between 1 week and 1 month after surgery. For the other two patients (6.5%), the resumption of postoperative antithrombotic regimens was delayed until 1 month later. Within 7 days after operation, reddish light hematuria was reported in 28 patients (90.3%). Crimson gross hematuria was reported in two patients, one of who was readmitted due to clot blockage of the catheter, and extra continuous bladder irrigation was performed to keep the catheter unobstructed. Within one month, 13 patients (41.9%) occasionally had reddish light hematuria. Only one patient with hematuria finally resolved after more than three months. None of the patients needed reoperation due to bleeding (Table 4). 

## 4. Discussion

BPH is a common urination disorder in middle-aged and elderly men, and it is one of the most common diseases in the clinical practice of urology around the world. Approximately 50% of men over 60 years old are troubled by BPH, and about 30% eventually need surgery [17,18]. Many elderly patients with cardiovascular and cerebrovascular diseases need long-term oral antithrombotic drugs [7]. Studies have shown that roughly 4% of patients who need TURP take anticoagulants orally for a long time [19], and a larger proportion of patients take antiplatelet drugs [20].

Hemorrhage is an unavoidable complication after TURP [21,22,23]. If antithrombotics are used continuously during the perioperative period, the risk of surgical hemorrhage will be enlarged considerably. However, discontinuation of antithrombotics increases the incidence of adverse cardiovascular and cerebrovascular events [24]. The European Association of Urology (EAU) recommends that the timing of preoperative discontinuation of antithrombotic drugs in non-extremely high-risk patients should be adjusted according to the type of antithrombotic drug, ranging from 12 h before surgery (e.g., for unfractionated heparin) to 5–7 days (e.g., for clopidogrel) [25]. Dimitropoulos K et al. [12] suggested that oral antithrombotic therapy should be discontinued 7–10 days before TURP in patients with low risk of cardiovascular events. There is relatively much literature in this field regarding the effect of preoperative discontinuation of antithrombotic drugs on postoperative bleeding after TURP [14,26,27,28]. However, these studies have mainly focused on a single drug (aspirin) and a single complication (postoperative bleeding). Our study shows that patients taking aspirin antithrombotic therapy alone account for about 60% of all antithrombotic patients, which means that 40% of patients may still be taking other antithrombotic therapies, who lack evidence-based guidance for stopping antithrombotic drugs before TURP. After detailed questioning during hospitalization and follow-up within three months after discharge, it became apparent that most of the 31 patients included in this study had been instructed to discontinue antithrombotic drugs within 7 to 14 days before surgery. Only one patient underwent bladder irrigation due to a clogged urinary catheter with blood clot postoperatively. The symptoms of hematuria lasted for 3 months at most postoperatively. No patient underwent reoperation because of hemorrhage within 3 months postoperatively. Therefore, the risk of postoperative bleeding may be acceptable if antithrombotic drugs are temporarily stopped before operation, but it is obvious that large-scale and high-level evidence is still needed to clarify this.

VTE is also a common and potentially fatal complication after operation. As the third leading cause of cardiovascular death, it has received more and more attention from clinicians in recent years [29], and it is also one of the common perioperative complications of urological surgery. However, SVT is more common in clinical practice and has always been regarded as a benign self-limiting disease. One study showed that 18.1% of SVT patients were combined with DVT and 6.9% were combined with PE [30]. Obviously, the risk of SVT cannot be ignored [31,32]. Therefore, patients who underwent TURP surgery with postoperative SVT were included in this study. A previous study pointed out that taking antithrombotic drugs for a long time will affect the balance of the anticoagulation/coagulation system of the body. The discontinuation of antithrombotic drugs before TURP may make the body become hypercoagulable in a short time, thus increasing the probability of VTE postoperatively [10]. In our study, among the 31 patients who took antithrombotic drugs for a long time, six patients (18.9%) suffered from SVT or DVT after operation, which is much higher than the incidence of VTE after TURP in the normal elderly population (0.5–1.4%) [30,33]. However, it is similar to a previous research result [34]. Taking antithrombotic drugs may be a high risk factor for VTE after TURP, and there may be two reasons. First, discontinuation of antithrombotic drugs disrupts the long-term balance of the coagulation/anticoagulation system. Second, patients with BPH are mostly old men, and it is undeniable that aging is a risk factor for VTE. 

In our study, although there were no serious complications such as pulmonary embolism, myocardial infarction, and cerebral infarction under active surveillance, the incidence of postoperative VTE in patients who stopped antithrombotic drugs before operation was significantly higher than that in the normal elderly population. Therefore, in clinical practice, it is still necessary to be highly alert to the risk of postoperative VTE in patients with previous antithrombotic therapy. Urologists need to raise awareness, actively monitor, and intervene in time.

Previous studies have focused on the effect of aspirin on bleeding after TURP, but as far as we know, there is a lack of research studying the effects of preoperative antithrombotic drug discontinuation on VTE after TURP. Our research takes both of them into account and expands the antithrombotic drugs from aspirin to various commonly used antithrombotic programs in clinics as well as exploring the discontinuation program of antithrombotic drugs with acceptable risk, which is more in line with the actual clinical situation. However, this study is a retrospective observational study, and it also has certain limitations. It cannot reveal the statistical difference in the incidence of VTE between patients treated with antithrombotic therapy and healthy elderly people, and it is not enough to draw a definite causal relationship. The sample size is also small, which may affect our analysis results. High-quality prospective multicenter studies are still needed for further analysis and confirmation in the future.

## 5. Conclusions

Under professional perioperative management, short-term preoperative discontinuation may help patients with antithrombotic therapy to obtain a relatively safe opportunity for TURP surgery. The risk of postoperative bleeding, VTE, and serious cardiovascular and cerebrovascular complications seems to be acceptable and controllable. It is essential for urologists to pay more attention to the perioperative management of these patients. However, this study is a single-center study with a small number of cases and thus needs further high-quality research results for more powerful verification.

## Figures and Tables

**Table 1 jcm-12-00417-t001:** Baseline information of patients and types, indications, and administration duration of antithrombotic drugs (n = 31).

	n (%)		n (%)
Age		Indications	
≥65 y	6 (19.4)	Coronary stent implantation	6 (19.4)
<65 y	25 (80.6)	Aortic valve replacement	2 (6.5)
Prostate volume		Mitral valve replacement	3 (9.7)
≤50 mL	13 (41.9)	Remote myocardial infarction	2 (6.5)
50–100 mL	16 (51.6)	Remote cerebral infarction	10 (32.3)
≥100 mL	2 (6.5)	Coronary heart disease	6 (19.4)
IPSS score		Atrial fibrillation	2 (6.5)
0–7	0 (0.0)	Drug information	
8–19	11 (35.5)	Aspirin	19 (61.3)
20–35	20 (64.5)	Clopidogrel	6 (19.4)
Operation history within 1 month	1 (3.2)	Aspirin and clopidogrel	4 (12.9)
Comorbidities		Warfarin	2 (6.5)
Hypertension	27 (87.1)	Medication duration	
Diabetes	5 (16.1)	≤1 year	3 (6.5)
Coronary heart disease	15 (48.4)	≤5 year	15 (48.4)
Apoplexy	16 (51.6)	≤10 year	11 (35.5)
Varicosity of lower limbs	1 (3.2)	>10 year	2 (6.5)

**Table 2 jcm-12-00417-t002:** Incidence of VTE after TURP (n = 31).

	n (%)
SVT	6 (19.4)
*SVT only*	*2 (6.5)*
*SVT combined with DVT*	*4 (12.9)*
*SVT combined with PE*	*0 (0.0)*
VTE (without SVT)	0 (0.0)
No VTE	25 (80.6)

**Table 3 jcm-12-00417-t003:** Perioperative conditions and clinical outcomes of TURP.

	n (%)
Preoperative withdrawal duration	
<1 week	3 (9.7)
1–2 week	24 (77.4)
>2 week	4 (12.9)
Pre-operation and post-withdrawal conditions	
New myocardial infarction	0 (0.0)
New cerebral infarction	0 (0.0)
Other new adverse events	0 (0.0)
Operation duration	
≤60 min	11 (35.5)
60–120 min	8 (25.8)
≥120 min	2 (6.5)
Intraoperative bleeding volume	
≤100 mL	28 (90.3)
100–400 mL	2 (6.5)
≥400 mL	1 (3.2)
Postoperative hospitalization days	
≤3 d	25 (80.6)
3–7 d	6 (19.4)
≥7 d	0 (0.0)
Time of postoperative antithrombotic recovery after operation	
<1 week	2 (6.5)
1 week–1 month after operation	27 (87.0)
>1 month after operation	2 (6.5)
Transfuse blood	0 (0.0)
Lower extremity discomfort	0 (0.0)
Postoperative MACEs	0 (0.0)
Cerebrovascular complications	0 (0.0)
Death	0 (0.0)

**Table 4 jcm-12-00417-t004:** Hemorrhage after TURP (n (%)).

	Time	7 Days after Operation	7 Days–1 Month after Operation	1–3 Months after Operation
Bleeding Conditions				
Reddish gross hematuria		28 (90.3)	13 (41.9)	1 (3.2)
Crimson gross hematuria		2 (6.5)	0 (0.0)	0 (0.0)
Bladder irrigation required		1 (3.2)	0 (0.0)	0 (0.0)
Reoperation required		0 (0.0)	0 (0.0)	0 (0.0)

## Data Availability

Not applicable.

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
