# Peer review of "Venous Thromboembolism and Bleeding after Transurethral Resection of the Prostate (TURP) in Patients with Preoperative Antithrombotic Therapy: A Single-Center Study from a Tertiary Hospital in China"

_jcm, 2023, doi:10.3390/jcm12020417_

Round 1

Reviewer 1 Report

Dear author 
-conclusion is not clear " In patients with antithrombotic therapy, the risk of post- 35 operative hemorrhage, VTE and serious cardiovascular and cerebrovascular complications after short-term 36 preoperative discontinuation seems to be acceptable. Urologists should pay more attention to the effect of 37 preoperative antithrombotic therapy on VTE and bleeding after TURP". 
Some punctuation errors are in the text that should be solved

Author Response

Point 1: conclusion is not clear " In patients with antithrombotic therapy, the risk of post- 35 operative hemorrhage, VTE and serious cardiovascular and cerebrovascular complications after short-term 36 preoperative discontinuation seems to be acceptable. Urologists should pay more attention to the effect of 37 preoperative antithrombotic therapy on VTE and bleeding after TURP".

Response 1: Thanks for the reviewer’s constructive suggestion. We have modified the conclusion section as follows. Short-term preoperative discontinuation may help patients with antithrombotic therapy to obtain a relatively safe opportunity for TURP surgery after professional evaluation of perioperative conditions. The risks of periop-erative bleeding, VTE and serious cardiovascular and cerebrovascular complications are relatively controllable. It is essential for urologists to pay more attention on the perioperative management of these patients. However, it needs further high-quality research results for more powerful verification. Please see the revised manuscript page 1, line 35-39 and page 1, line 255-261 .

Point 2: Some punctuation errors are in the text that should be solved.

Response 2: Thanks for the suggestion. We are very sorry for our negligence. We have read and corrected our revised manuscript carefully for the punctuation errors.

Reviewer 2 Report

Reviewer´s comments on Venous thromboembolism and bleeding after transurethral resection of the prostate (TURP) in patients with preoperative antithrombotic therapy: A single-center study from a tertiary hospital in China

Zhongyi Li et al evaluated the effect of a history of preoperative anticoagulant therapy on the occurrence of complications after transurethral resection of the prostate (venous thromboembolism and bleeding).

This manuscript provides original research by clearly describing the hypothesis, study and used methods.

Overall I find nothing to declare in respect of appropriateness of context and purpose of study reflected by the abstract and the introduction. The methods used are valid, underlined by previous publications and the results are correctly presented in good quality tables. The discussion and conclusions are supported by the data.

Finally, I come to the conclusion that the authors provide interesting data, congratulate them for their good work.

Best regards

Author Response

Point 1:  Zhongyi Li et al evaluated the effect of a history of preoperative anticoagulant therapy on the occurrence of complications after transurethral resection of the prostate (venous thromboembolism and bleeding).This manuscript provides original research by clearly describing the hypothesis, study and used methods. Overall I find nothing to declare in respect of appropriateness of context and purpose of study reflected by the abstract and the introduction. The methods used are valid, underlined by previous publications and the results are correctly presented in good quality tables. The discussion and conclusions are supported by the data. Finally, I come to the conclusion that the authors provide interesting data, congratulate them for their good work..

Response 1: We are very pleased to receive your valuable comments and acknowledgement of our work. Thank you.
